# Myocardial Work by Echocardiography: Principles and Applications in Clinical Practice

**DOI:** 10.3390/jcm10194521

**Published:** 2021-09-29

**Authors:** Federica Ilardi, Antonello D’Andrea, Flavio D’Ascenzi, Francesco Bandera, Giovanni Benfari, Roberta Esposito, Alessandro Malagoli, Giulia Elena Mandoli, Ciro Santoro, Vincenzo Russo, Mario Crisci, Giovanni Esposito, Matteo Cameli

**Affiliations:** 1Department of Advanced Biomedical Sciences, Federico II University Hospital, 80131 Naples, Italy; roberta.esposito6@gmail.com (R.E.); cirohsantoro@gmail.com (C.S.); espogiov@unina.it (G.E.); 2Mediterranea Cardiocentro, 80122 Naples, Italy; 3Unit of Cardiology, Department of Traslational Medical Sciences, University of Campania “Luigi Vanvitelli”, Monaldi Hospital, 80131 Naples, Italy; antonellodandrea@libero.it (A.D.); v.p.russo@libero.it (V.R.); 4Unit of Cardiology and Intensive Coronary Care, “Umberto I” Hospital, 84014 Nocera Inferiore, Italy; 5Department of Medical Biotechnologies, Division of Cardiology, University of Siena, 53100 Siena, Italy; flavio.dascenzi@unisi.it (F.D.); giulia_elena@hotmail.it (G.E.M.); matteo.cameli@yahoo.com (M.C.); 6Department of Biomedical Sciences for Health, University of Milan, 20122 Milan, Italy; francescobandera@gmail.com; 7Section of Cardiology, Department of Medicine, University of Verona, 37132 Verona, Italy; giovanni.benfari@gmail.com; 8Division of Cardiology, Nephro-Cardiovascular Department, Baggiovara Hospital, University of Modena and Reggio Emilia, 41126 Modena, Italy; ale.malagoli@gmail.com; 9Department of Cardiology, Division of Interventional Cardiology, Monaldi Hospital, 80131 Naples, Italy; mario.crisci1984@gmail.com

**Keywords:** myocardial work, pressure-strain loops, strain, speckle tracking, myocardial function

## Abstract

Left ventricular (LV) global longitudinal strain (GLS) has established itself in the last decade as a reliable, more objective method for the evaluation of LV systolic function, able to detect subtle abnormalities in LV contraction even in the presence of preserved ejection fraction (EF). However, recent studies have demonstrated that GLS, similar to LV EF, has important load dependency. Non-invasive myocardial work (MW) quantification has emerged in the last years as an alternative tool for myocardial function assessment. This new method, incorporating measurement of strain and LV pressure, has shown to overcome GLS and LV EF limitations and provide a loading-independent evaluation of myocardial performance. The presence of a commercially available echocardiographic software for the non-invasive MW calculation has allowed the application of this new method in different settings. This review sought to provide an overview on the current knowledge of non-invasive MW estimation, showing its potential applications and possible added value in clinical practice.

## 1. Introduction

Left ventricular ejection fraction (LVEF) is considered the first-line tool for left ventricular (LV) systolic function description. Due to its wide diffusion, simple estimation, and extensive use in clinical trials and guideline recommendations for various diseases, LVEF is the most commonly used surrogate marker of LV function in many cardiac diseases. Despite its widespread use, LVEF has important limitations [1]: as a volume-derived index, it relies on geometric assumptions and is extremely load-dependent, thus leading to considerable loss of reproducibility [2]; it may be influenced by changes in geometry (e.g., hypertrophic or dilated LV) and does not reflect the true LV contractility; it is poorly sensitive in detecting declining ventricular function [3]. All these factors prompted the search for new indices of myocardial function. Over the past decade, LV global longitudinal strain (GLS) has established itself as a viable alternative for the evaluation of LV systolic function [4]. Speckle-tracking-derived longitudinal strain is a semiautomated method that guarantees a lower inter-and intra-observer variability in the analysis of myocardial contraction [5]. Thus, the more objective quantification of LV systolic function and the higher sensibility to detect more subtle abnormalities in LV contraction even when LVEF is normal represent the main advantages of GLS, that allowed an exponential increase of its application in different clinical setting [6]. However, several studies have demonstrated that GLS, similarly to LVEF, has important load dependency, hence it is affected in condition of elevated pre- or after-load [7].

In the recent years, myocardial work (MW) has emerged as an alternative tool for myocardial function assessment. This new parameter derives from GLS, with the advantage to incorporate information on afterload, through interpretation of strain in relation to dynamic non-invasive LV pressure. The presence of a commercially available echocardiographic software for the non-invasive MW calculation has allowed the application of this new method in different settings.

This review sought to provide an overview on the current knowledge of non-invasive MW estimation, showing its potential applications in clinical practice.

## 2. Evaluation of Myocardial Work from Estimated LV Pressure Curves

In physiologic conditions, myocardial contraction provides energy necessary to eject blood out of the ventricles and to circulate through the body. Actually, the ratio between the mechanical energy developed by the myocardium and imparted to the blood and the total energy consumed depends upon the loading conditions. As previously demonstrated [8], the analysis of pressure-volume loops allows the measurement of the energy imparted to the blood, as the area of the loop represents the stroke work [9]. Subsequently, Suga et al. [10] showed in experimental study that pressure-volume area serves as a predictor of cardiac oxygen consumption. This result was also confirmed by Takaota and colleagues in clinical setting [11]. Based on the same principle, pressure-dimension loops were used to estimate regional LV function and segmental work, that appeared to be particularly useful in the study of synchrony or dyssynchrony of contraction [12]. However, the need for invasive measures to estimate pressure-dimension loops area has limited until now the use of this index. Recently, Russell et al. introduced a non-invasive method for assessing regional MW, by the analysis of LV pressure-strain loop (PSL) [13]. In this new method, LV pressure curve was derived from non-invasively acquired brachial artery cuff pressure and generated by adjusting the profile of a reference LV pressure curve according to the duration of the isovolumic and ejection phases, as defined by timing of aortic and mitral valve events by echocardiography. LV PSL represented, thus, the result of the integration of the estimated LV pressure curve with strain by speckle tracking echocardiography. In their experimental study, Russell and colleague demonstrated a good agreement between the proposed method and the loop area by invasive PSLs [13]. Moreover, non-invasive PSL area showed a strong correlation with regional glucose metabolism using 18-fluorodeoxyglucose-positron emission tomography, therefore supporting the use of this method as an index of regional MW. Based on these encouraging results from experimental studies [13,14], the method has been included in an echocardiographic software. A normalized pressure curve was obtained by pooling invasive pressure measurements from a number of patients with different pathologies, to make the method applicable also in several pathological settings, and then normalized to equal durations of isovolumetric contraction, ejection and isovolumetric relaxation, as well as peak pressure.

Certainly, the estimation of pressure curve from the arterial systolic pressure, measured by a brachial cuff, holds the chance to make MW measurement easily achievable. However, an experimental study aimed at verifying the accuracy of the algorithm proposed by Russell et al. in different hemodynamic conditions, showed that the estimation of pressure curve is imperfect, since the accuracy varies along the cardiac cycle [14]. Moreover, the imprecision in pressure prediction grows when the arterial pressure is high and in presence of brachial vascular disease. Paradoxically, despite the inaccuracy in LV pressure estimation, authors demonstrated a precise MW estimation, probably related to the temporal integration that induces a smoothing of the differences between measured and estimated work [14]. This latest study therefore confirms the reliability of the method and its potential value in pathological conditions.

## 3. Quantification of Cardiac Work: Global Work Index, Constructive Work, Wasted Work and Work Efficiency

In the currently available echocardiographic software, after calculating GLS in 2-, 4-chamber and long-axis apical views, values of brachial blood pressure and time of valvular events are needed to derive PSLs. Valvular event times are set by pulse-wave Doppler recordings at mitral valve and aortic valve level, and then confirmed by 2D evaluation of the apical long-axis view (Figure 1).

The area of PSL represents approximately the MW and is calculated by computing the rate of segmental shortening, obtained by differentiating the strain curve, and multiplying this value with instantaneous LV pressure. This product is a measure of instantaneous power, which is integrated over time during the cardiac cycle to obtain MW, expressed in mmHg%. In addition to MW index (work evaluated from mitral valve closure to mitral valve opening), segments were analyzed for wasted work (WW) and constructive work (CW), with global values determined as the averages of all segmental values.

Table 1 summarizes the determinants of MW indices generated by the software and reports normal reference values according to age and gender, calculated from a population of 226 healthy subjects [15]. Interestingly, absence of strong dependence of MW on age and gender was described. Although an age-related increase in global work index (GWI) and global constructive work (GCW) was observed in the female population, it seemed to be mainly related to the increased blood pressure [15].

## 4. Clinical Applications

In general, MW may allow an in-depth evaluation of myocardial systolic performance across a broad range of physiologic and pathologic conditions beyond traditional echocardiographic techniques. While MW indices, mainly GWI and GCW, have shown good correlation with EF and strain parameters [16], the prospect of providing incremental information, not affected by loading conditions, and more insight into myocardial energetics have made MW application particularly useful in various clinical settings (Table 2).

### 4.1. Heart Failure and Cardiac Resynchronization

The first and more promising application of segmental MW is the prediction of therapeutic effects and outcomes in heart failure (HF) patients undergoing cardiac resynchronization therapy (CRT). Currently, CRT plays a key role in the treatment of symptomatic HF patients with LVEF ≤35% and wide QRS complex [40]. However, about 30% of patients fail to show a substantial benefit with CRT. In recent years, several efforts have been made to identify echocardiographic petameters able to predict CRT response, without promising results. Indeed, in a multicenter trial enrolling 498 patients with standard CRT indications, none of the 12 conventional and tissue Doppler-based echocardiographic indices of dyssynchrony were shown to be a reliable predictor of CRT response [41]. Although longitudinal, circumferential, radial strain [42,43,44] and strain rate [45] have shown good accuracy in discriminating patients that could benefit from CRT implantation, prospective randomized trials using STE are still lacking.

Some authors have supposed that the assessment of residual myocardial contractility in patients with dyssynchrony could play a role in the prediction of LV functional restoration after CRT [42,46]. In this context, MW seems to be able to identify patients suitable for myocardial improvement after resynchronization. A recent study showed that in CRT-responder global wasted work (GWW) and the average WW measured at the septum are higher compared to non-responders, but after CRT implantation both indices significantly reduce, approaching values of a normal heart [21]. Moreover, when combining the degree of septal WW with LV wall motion score, which was used to identify transmural myocardial scar, the prediction of response to CRT is even better: the area under the curve (AUC) for the combined parameters was 0.86, compared to an AUC of 0.80 and 0.63, respectively, for septal WW and wall motion score index alone. This result would suggest that a combined approach may be useful [21]. In a larger prospective multicenter study enrolling 200 CRT recipients, work difference between septum and lateral wall has demonstrated to be a good predictor of CRT response, with an AUC of 0.77 [47]. Interestingly, the combination of work difference with septal viability evaluated with cardiac magnetic resonance increased AUC to 0.88 (sensitivity of 86%, specificity of 84%), demonstrating to be superior to other electrocardiographic and echocardiographic parameters in the prediction of response to CRT therapy. Additionally, the redistribution of MW across septal and lateral wall after CRT has demonstrated to be a useful index to predict long-term reverse remodeling [48]. In a cohort of 97 patients, Galli et al. demonstrated that GCW is an independent predictor of CRT response at 6-months follow-up and is significantly associated with the entity of myocardial remodeling in both ischemic and non-ischemic patients [20]. Despite that AUC’s comparison did not reveal a superiority of GCW over other echocardiographic parameters (GCW >1057 mmHg%: AUC = 0.719, septal flash: AUC = 0.721, difference between areas: 0.002, *p* = 0.92), a GCW <1057 mmHg% identified 85% of non-responders with a positive predictive value of 88%. Taken together, all these data have been convincing about the role of MW indices, mostly segmental but also global, in the prediction of CRT therapy response and LV remodeling. Accordingly, the evaluation of MW indices, especially wasted and constructive work, should be routinely performed in this setting of patients. Of course, available evidence on these echocardiographic parameters is not robust enough to allow them to guide treatment alone, but an integrating approach, which combines clinical, electrocardiographic and advanced echocardiographic parameters, may help in the right selection of CRT candidates.

In patients with HF and reduced EF (HFrEF), the treatment with Sacubitril/Valsartan has also shown to significantly increase GCW after 6-months follow-up, while the improvement of global work efficiency (GWE) becomes evident at 12-months follow-up with respect to baseline [17]. Thus, GCW improvement seems to be a sensible index of LV wall stress reduction [49] and myocardial metabolism increase [50] induced by Sacubitril/Valsartan. Conversely, a baseline value of GCW <910 mmHg% identifies patients at high risk of major cardiovascular events [17]. Another potential role of GCW in the assessment of LV contractile reserve has been investigated in patients with HF and preserved EF (HFpEF) which is known to be characterized by exercise intolerance and poor peripheral oxygen extraction [51]. In a study enrolling HFpEF patients in treatment with spironolactone, the exertional increase in GCW has shown to be independently associated with improvement in exercise capacity at 6-months follow-up, while a spironolactone-related increase of GLS has not been described [19]. Recently, in a larger study, D’Andrea and colleagues provided further insights on the potential role of MW indices in the characterization of HFpEF patients [52]. More precisely, increased value of GWW at rest and during effort, despite normal EF and GLS, were detected in HF patients compared to controls, as a sign of subclinical impairment of systolic and diastolic function typical of this setting of patients. In addition, MWE at rest was closely related to exercise capacity, pulmonary congestion, and reduced LV contractile reserve during physical effort [52]. Certainly, MW provides an additional tool in the characterization of myocardial performance, particularly useful in HF patients with subclinical or evident LV dysfunction. However, results on the prognostic role of MW indices in this setting are still poor and cannot provide safe conclusions about the routine use of MW in HF patients.

### 4.2. Coronary Artery Disease

Non-invasive detection of early ischemia in patients with significant coronary artery disease (CAD) and normal resting systolic function is challenging and still being investigated [53]. In previous studies, GLS has demonstrated to be a strong predictor of stable ischemic cardiopathy even in the absence of wall motion abnormalities [54]. However, there is still no consensus on the optimal GLS diagnostic cutoff value, which varies significantly across the studies, related to clinical characteristics, afterload dependence or intervendor differences. Moreover, contractile patterns of ischemic myocardium are strongly influenced by loading condition with immediate changes from hypokinesis to dyskinesis following acute elevation in afterload [55]. MW estimation has shown to successfully overcome this limitation, and to provide diagnostic and prognostic information, both in chronic and acute setting. Edwards et al. demonstrated that in patients with suspected CAD and normal systolic function, GWI, GCW and GWE are significantly reduced in the presence of obstructive disease, while GWW slightly increases [22]. They reported that a value of 1810 mmHg% for GWI is able to detect CAD with a positive predictive value of 95%. Interestingly, GWI has shown to be superior to GLS to predict significant CAD, with an AUC of 0.786 for the GWI compared to 0.693 for GLS [22].

In non-ST-segment acute coronary syndrome, regional MW index has shown to be superior to all other echocardiographic parameters (GLS, LVEF, etc.) to identify acute coronary artery occlusion [23]. The presence of ≥4 adjacent segments with MW index <1700 mmHg% has shown 81% of sensitivity and 82% of specificity in detecting coronary occlusion, with a negative predictive value of 94%, demonstrating to be superior to functional risk area measured with strain (sensitivity 78%, specificity 65%, negative predictive value 91%). The superiority of GWI at identifying patients with acute coronary occlusion was evident also compared to LVEF (sensitivity 70 vs. 63%, specificity 82 vs. 62%, negative predictive value 91 vs. 86%, respectively) [54]. In a population of 93 patients with anterior ST-elevation myocardial infarction (STEMI), all MW indices appeared reduced in the left descending coronary artery territory, but significantly improved at 3-months follow-up in those patients with LV recovery. Among the different indices, CW has demonstrated an independent and incremental value in predicting segmental and global LV recovery, over standard (LVEF) and advanced (GLS) parameters, and in-hospital complications, such as HF or LV apical thrombus [26].

More reduced GWI, GCW, and GWE and more increased GWW, instead, have been detected in STEMI patients who developed LV ischemic remodeling at 3-months follow-up [25]. Consistent with these results, El Mahdiui et al. has shown that GWE is lower in patients who have undergone primary percutaneous coronary intervention for STEMI compared with healthy controls and those with cardiovascular risk factors, and even more impaired in presence of HFrEF [24]. These findings suggest that MW impairment is the expression of altered (persistent anaerobic) energy metabolism that occurs in remodeled myocardium [56] (Figure 2).

### 4.3. Hypertension and Diabetes Mellitus

Arterial hypertension is an ideal model for assessing the changes in myocardial deformation and performance related to the pressure overload and the development of LV concentric geometry. Indeed, arterial pressure is one of the most important independent predictors of an accelerated decline in GLS during a follow-up period [57,58]. Moreover, LV function investigation in hypertensive patients through MW allows to account for acute loading over the course of the cardiac cycle, potentially separating influences of blood pressure at the time of observation from the impact of chronic remodeling on regional deformation. Chan et al. described in a small group of patients with varying degrees of hypertension increased values of GWI and GCW, mainly detectable in those with uncontrolled hypertension, which reflect the enhanced contractility of the LV, that pumps with higher levels against elevated pressures [27]. These data have been further confirmed by Tadic et al., who have also shown in hypertensive patients an additional impact on GCW of type-2 diabetes mellitus [28]. An analysis on 170 hypertensive patients also revealed an apex-to-base gradient in the distribution of MW, which reflected an impairment of the basal segments, compensated by the apical region, more pronounced in those with basal septal hypertrophy [59]. Thus, in the context of hypertensive patients MW seems to be more sensitive. None of these studies were designed to prove the superiority of MW to GLS. Larger scale studies are needed to establish MW indices clinical utility and prognostic impact on cardiovascular outcome in this setting of patients.

### 4.4. Cardiomyopathies

In patients with non-obstructive hypertrophic cardiomyopathy (HCM), GWI, GCW and GWE have shown to be significantly impaired compared to controls with similar LVEF [29,30], thus reflecting fibers disarray and metabolic impairment, with an increase of GWW. In particular, GCW correlated with LV diastolic dysfunction, maximum LV wall thickness and QRS duration [31]. Interestingly, GCW emerged as the only predictor of LV fibrosis assessed by late gadolinium enhancement [29], and values of GCW <1730 mmHg% have been associated with worse long-term outcome [30]. The application of MW indices in HCM patients seems to be particularly promising, as afterload might change with medication use or geometric changes and increase of wall thickness over time. However, the additive value of these new parameters over the traditional ones is still object of study.

Additionally, in a subgroup of patients with dilated cardiomyopathy (DCM), Chan et al. described a significant reduction of GWI, GCW and GWE, with an increase of GWW, which reflected an important impairment of cardiomyocytes contractile performance within DCM patients, either ischemic or non-ischemic [27]. After, Cui et al. provided further insights in non-ischemic DCM patients, showing that after 6 months of therapy, a 6-min walking distance increase is accompanied by a significant improvement of GWI values, without changes in LVEF and GLS [31]. This result would suggest more sensitivity of MW indices in detecting functional cardiac improvement and evaluating the effectiveness of therapy. Another potential clinical application of MW indices is in the prediction of major cardiovascular events in cardiac amyloidosis (CA) [33,34]. It has been shown that in a CA population, GWI and GWE are impaired despite a preserved LVEF, with a more pronounced reduction in the basal segments, and a consequent alteration in the average apical-to-basal segmental ratios, compared to controls [32] (Figure 3A,B). These indices of LV myocardial performance have shown a good correlation with N-terminal pro b-type natriuretic peptide (NT-proBNP), estimated glomerular filtration rate (eGFR), troponin and peak oxygen consumption, which are known prognostic parameters [33]. Interestingly, in a population of 100 patients with CA, GWI was superior to GLS to predict all-cause death (hazard ratio [HR] 2.6 (95% CI: 1.2–5.5), AUC 0.68, sensitivity 66%, specificity 63%, vs. HR 1.7 (95%CI: 0.8–2.5), AUC 0.65, sensitivity 55%, specificity 55%, respectively) [34]. Furthermore, the combination of GWI and apical-to basal segmental work ratio demonstrated to be even stronger in the prediction of major adverse cardiovascular events (MACE), defined as composite of rehospitalization due to HF and all-cause death, and mortality [34].

### 4.5. Athlete’s Heart

The analysis of MW in endurance athletes represents an interesting application to evaluate LV myocardial deformation and contractile reserve in a less loading-dependent manner (Figure 3C,D). A large study on 350 athletes compared to 150 controls demonstrated that, at rest, LV adaptation results in preserved GWE and GWW despite a reduction of GLS [35]. Moreover, GWE at rest is able to predict functional capacity and pulmonary or hemodynamic congestion measured at peak effort, better than LVEF. Evaluation of MW indices in 24 half-marathon runners before, immediately post, and 72 hours after a marathon confirmed that GWE and GWW do not change significantly after the competition, but an increase in GWI has been observed in a subgroup of athletes with higher BNP values and higher heart rate [60]. Authors suggested that increased value of GWI post-marathon would reflect an early manifestation of myocardial stress with increased heart rate, which may be a precursor to myocardial fatigue.

## 5. Emerging Areas of Application

### 5.1. Stress Echocardiography

The assessment of myocardial performance during exercise stress echocardiography through non-invasive MW estimation has been recently proposed as a valuable method to overcome the change in loading condition occurring during the test. In their report, Halabi et al. showed that MW is feasible during stress echocardiography [36], and GWI measured at the peak stress was correlated with functional capacity, a well-known predictor of mortality in healthy subjects [61,62]. Mansour and colleagues have provided more insights, describing an increase of MW indices (GCW, GWI and GWW) in patients with uncontrolled peak systolic blood pressure (SBP >180 mmHg), with a relatively preserved GWE and without significant changes in GLS [37]. Moreover, an elevated peak SPB was strongly associated with abnormal GWW.

Further and larger studies are needed to better understand the additive value and potential usefulness of the application of this new tool during stress echocardiography on the measurement of myocardial reserve and on the prediction of myocardial ischemia.

### 5.2. Valvular Heart Disease

In actuality, the chronic overload that characterizes aortic regurgitation is responsible for LV remodeling, starting from eccentric hypertrophy, progressive dilation, and LV systolic dysfunction. In this context, MW analysis may allow a more reliable quantification of myocardial performance corrected by overload, permitting to individuate patients with incipient LV systolic decompensation. Indeed, in a study conducted in patients with asymptomatic severe aortic regurgitation undergoing physical effort, D’Andrea et al. demonstrated that baseline GLS and MWE are significantly correlated with functional capacity, LV filling pressure, and pulmonary congestion during effort [38].

Since MW measurement is based on the estimation of non-invasive LV pressure from SBP measured with a cuff manometer, its evaluation was not recommended in pathologic conditions such as aortic stenosis (AS), in which SBP is not representative of LV peak systolic pressure owing to the fixed obstruction of a stenotic valve. However, the study of myocardial energetics through MW would be particularly useful in this setting, since LVEF remains preserved until late stages of the aortic disease, while GLS, which is a well-known predictor of worse prognosis, has shown to have important load dependence [63,64]. Recently, Jain et al. have proposed to use the sum of transaortic mean gradient and SBP as an estimation of LV peak systolic pressure in a population of severe AS patients undergoing transcatheter aortic valve replacement (TAVR) [39]. They have found a high correlation between LV systolic pressure invasively measured and the one estimated non-invasively, which were used for calculations of MW. Comparing MW indices pre- and post-TAVR, a significant reduction of GWI and GCW has been observed that has been attributed to the immediate relief of the heightened oxygen demand related to the increased afterload [39] (Figure 4). Fortuni et al. have further confirmed the excellent agreement between LV MW indices calculated with invasive versus echocardiography-derived LV systolic pressures in a population of 120 patients with severe AS [18]. Furthermore, they found that lower values of echocardiography-derived GWI and GCW are independently associated with NYHA III-IV HF symptoms, in contrast to LV GLS. According to these results, the proposed method of MW correction by adding transaortic mean gradient to SBP seems to be feasible and reliable. However, its validation in larger AS populations undergoing TAVI is necessary to allow its introduction in routine practice.

### 5.3. Right Ventricular Myocardial Work

As previously described, non-invasive MW indices are currently measured with a software validated for the measurement of LV MW. Recently, a proof-of-concept study showed the feasibility of right ventricle (RV) MW indices measurement in a small population of HFrEF [65]. In addition to the reduction of RV GWI, GCW, GWE and increase of GWW in HF patients compared to controls, they demonstrated that RV GCW is the only echocardiographic parameter associated with invasively measured stroke volume [65]. Therefore, MW seems to also be able to provide an all-around evaluation of the RV function, integrating information on contractility, dyssynchrony and pulmonary pressure, and overcoming the limitations of other standard parameters such as TAPSE, RV GLS or fractional area change [66]. Further and larger studies are needed to validate the application of MW software in studying the right chamber and extending its applicability in other clinical settings.

## 6. Limitations

Since MW calculation is a speckle-tracking–derived measurement, it has inherited technical limitations, partly inherent to echocardiography: poor image quality that does not allow correct endocardial border delineation, and difficult to scan anatomy often leading to foreshortened ventricles, which impact on longitudinal strain (and consequently MW indices) measurements in the apex. Moreover, although having incorporated LV pressure measure makes them less sensitive to LV afterload than GLS, MW indices cannot be considered load-independent, as their values are derived by strain measurements. Considering the difficulty in obtaining strain traces and MW estimation in cases of significant variability, in most of the studies patients with atrial fibrillation have been excluded. Actually, only one system currently provides software to calculate MW, limiting the applicability of this method.

## 7. Conclusions

Validation studies have demonstrated that the non-invasive estimation of MW indices obtained from LV PSL strongly correlates with invasive measurement of stroke work and with cardiac metabolism. This has allowed a broad application of MW measurement in several clinical settings.

The main advantage of this new diagnostic tool is that, integrating stain measurement with pressure, it yields a more objective evaluation of ventricular function, which incorporates loading conditions and overcomes EF and GLS limitations. Thus, MW assessment may become particularly useful in situations to elucidate if the reduced contraction is due to increased afterload (such as arterial hypertension) or attenuated contractility. Furthermore, the ability to provide more insight into segmental and global myocardial energetics has opened new horizons in the study of cardiomyopathies and in the prediction of response to therapy.

Within the last years, a growing body of evidence has accumulated, showing the good feasibility and reproducibility of MW, and supporting its use in several clinical applications. However, multicenter well-designed studies for the validation of this technique in large populations are needed, to definitively attest its added value and in order to include MW indices in the routine echocardiographic evaluation.

## Figures and Tables

**Figure 1 jcm-10-04521-f001:**
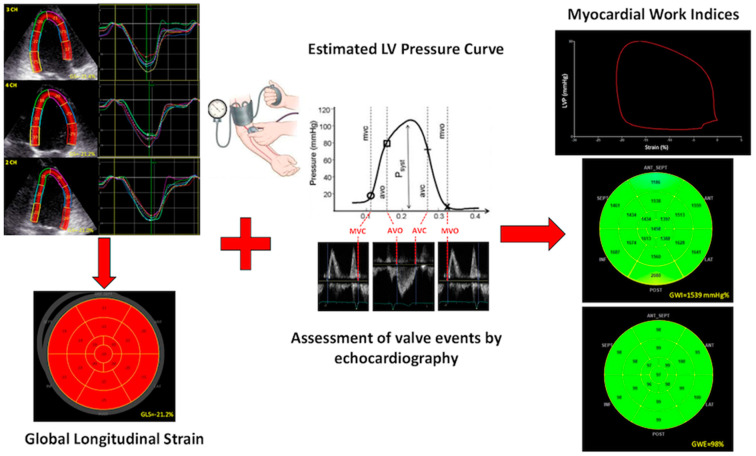
LV global longitudinal strain data, measured using the R-wave onset in the electrocardiogram as a common time reference (left panel), are combined with estimated LV pressure curve. After measuring peak arterial pressure with a cuff manometer, an empiric, normalized reference curve is adjusted according to LV duration of the isovolumetric and ejection phases, defined by timing of aortic and mitral valve events by echocardiography (central panel). On the right panel, representative trace showing LV pressure-strain loop (up), 17-segment bull’s-eye representation of MW index (middle) and MW efficiency (bottom). AVC = aortic valve closure; AVO = aortic valve opening; GWI = global work index; GWE = global work efficiency; LV = left ventricular; LVP = left ventricular pressure; MVC = mitral valve closure; MWO = mitral valve opening.

**Figure 2 jcm-10-04521-f002:**
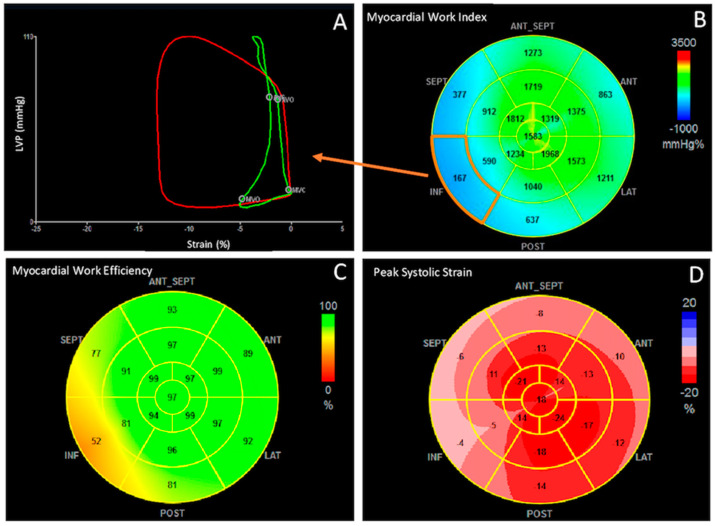
Myocardial work indices in a patient with inferior ST-elevation myocardial infarction. Pressure-strain loop of the basal segment of the inferior wall showed post-systolic shortening, early marker of ischemia, which translate into an important decrease of MW index (**A**,**B**), global work efficiency (**C**) and global longitudinal strain (**D**) in the territories supplied by the obstructed right coronary artery. LVP = left ventricular pressure.

**Figure 3 jcm-10-04521-f003:**
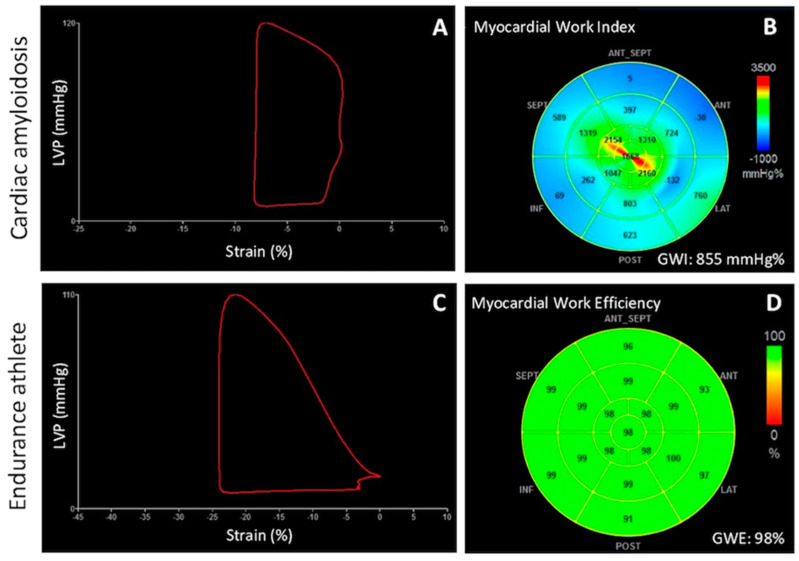
Pressure-strain loop (**A**) and 17-segment bull’s-eye representation of GWI (**B**) in a patient with cardiac amyloidosis, showing a reduced loop area, representing a reduced stroke work developed by the ventricle, and significant impairment of MW index in the basal segments. In endurance athletes MWE is an early predictor of physiologic remodeling in baseline examination (**C**,**D**). GWI = global work index; GWE = global work efficiency; LVP = left ventricular pressure.

**Figure 4 jcm-10-04521-f004:**
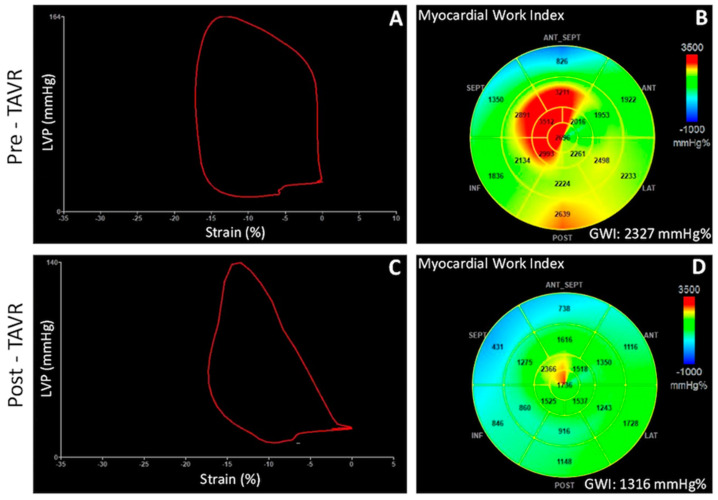
Pressure-strain loop (**A**,**C)** and 17-segment bull’s-eye representation of global work index (**B**,**D**) in a patient with severe aortic stenosis before and after transcatheter aortic valve implantation, demonstrating a significant reduction of stroke work, and therefore GWI, as a consequence of the reduced afterload post valve replacement. GWI = global work index; LVP = left ventricular pressure; TAVR = transcatheter aortic valve replacement.

**Table 1 jcm-10-04521-t001:** Determinants of myocardial work indices and reference values according to gender.

Parameter	Determinants	Reference Values [15]		
		Total	Male	Female
**GWI** **(mmHg%)**	Amount of myocardial work performed by the left ventricle during systole => area of PSL from mitral valve closure to mitral valve opening	1292–2505	1270–2428	1310–2538
**GCW** **(mmHg%)**	Positive work performed in systole (shortening) + Negative work performed in isovolumetric relaxation (lengthening)	1582–2881	1650–2807	1543–2924
**GWW** **(mmHg%)**	Negative work performed in systole (lengthening) + Positive work performed in isovolumetric relaxation (shortening)	226 ± 28 ^a^	238 ± 33 ^a^	239 ± 39 ^a^
**GWE** **(%)**	Percentage (0–100%) of constructive work over total work => Constructive work/(constructive work + wasted work)	91 ± 0.8 ^b^	90 ± 1.6 ^b^	91 ± 1 ^b^

Data are expressed as 95% confidence interval or limits of normality ± standard error ^a, b^. ^a^ Highest expected value. ^b^ Lowest expected value. GCW = global constructive work; GWE = global work efficiency; GWI = global work index; GWW = global wasted work; PSL= pressure-strain loop.

**Table 2 jcm-10-04521-t002:** Main clinical applications of MW indices.

Clinical Setting	Diagnostic Role	Prognostic Role
Heart failure and reduced EF (HRrEF)	- GCW and GWE increase associated with sacubitril/valsartan treatment [17] - Reduction of RV GWI, GCW, GWE and increased RV GWW [18] - Association of RV GCW with invasively measured stroke volume [18]	- GCW <910 mmHg is predictor of MACE [17]
Heart failure and preserved EF (HRpEF)		- Exertional increase in GCW predicts LV systolic reserve response to spironolactone [19]
Cardiac resynchronization		- GCW>1057 mmHg% is predictor of CRT-positive response [20] - WW at the septum is predictor of CRT-positive response [21]
Coronary artery disease (CAD)	- Reduction of GWI, GCW and GWE in presence of significant CAD but normal LVEF and wall motion [22] - Regional WI<1700 mmHg% in more than 4 adjacent segments detects acute coronary occlusion in NSTE-ACS [23] - Lower GWE in patients undergoing primary PCI for STEMI [24] - More impaired GWI, GCW, GWE and increased GWW in postinfarction remodeling [25]	- GWI<1810 mmHg% predicts significant CAD [22] - GCW predicts global and regional LV recovery after STEMI [26]
Hypertension (HTN) and diabetes mellitus (DM)	- Elevation of GWI and GCW in uncontrolled HTN (Grade 2/3) [27] - Progressive increase of GWI and GCW in HTN patients with concomitant DM [28] - Independent association between GCW or GWI with systolic blood pressure and DM [28]	
Non-obstructive hypertrophic cardiomyopathy (HCM)	- Impairment of GCW, GWI and GWE [29,30] - Increase of GWW - GCW correlations with maximum LV wall thickness, diastolic function, and QRS duration [30] - Segmental CW impairment among different HCM phenotypes [30]	- GCW<1730 mmHg% is predictor of worse long-term outcome [30] - GCW<1623 mmHg% is predictor of LV fibrosis [29]
Dilated cardiomyopathy (DCM)	- Impairment of GCW, GWI, GWE and increase of GWW in both ischemic and non-ischemic DCM [27]	- Improvement of GWI and 6-min walking test after 6-month therapy [31]
Cardiac amyloidosis	- GWI and GWE impairment, mainly in the basal segments, despite preserved LVEF [32] - GWI and GWE correlate with NT-proBNP, eGFR, troponin and peak oxygen consumption [33]	- GWI<1043 mmHg% predictor of MACE [34] - GWI<1039 mmHg% predictor of all-cause death - Combination of GWI and apical-to-basal segmental work ratio independently associated with MACE [34]
Athlete’s heart	- Preserved GWW and GWE, despite reduced GLS, expression of LV physiologic adaptation [35] - GWI increase post-marathon correlated to higher BNP values heart rate [35]	- GWE predicts maximal watts, peakVO_2_, LV E/e’ and numbers of B-lines at peak effort [35]
Stress echocardiography	- Weak correlation between peak GWI and functional capacity [36] - Peak stress GWI, GCW and GWI increase when SBP >180 mmHg [37]	
Aortic regurgitation	- MWE impairment in severe disease [38] - MWE at baseline correlates with peak effort watts, peak VO_2_, LV E/e’ and numbers of B-lines [38]	- GWE predicts contractile reserve [38]
Aortic stenosis	- Reduced GWI and GCW in severe AS patients with HF symptoms [18] - Reduction of GWI and GCW post TAVR [39]	

AS = aortic stenosis; BNP = brain natriuretic peptide; CAD = coronary artery disease; DCM = dilated cardiomyopathy; DM = diabetes mellitus; E/e’ = transmitral Doppler E wave velocity/mean e’ velocity; GCW = global constructive work; GFR = glomerular filtration rate; GLS = global longitudinal strain; GWE = global work efficiency; GWI = global work index; GWW = global wasted work; HCM = hypertrophic cardiomyopathy; HF = heart failure; HTN = hypertension; LV = left ventricular; NSTE-ACS = non-ST-segment acute coronary syndrome; PCI = percutaneous coronary intervention; RV = right ventricular; SBP = systolic blood pressure; STEMI = ST-elevation myocardial infarction; TAVR = transcatheter aortic valve replacement; WI = work index; WW = wasted work; NT-proBNP = N-terminal pro b-type natriuretic peptide; eGFR = estimated glomerular filtration rate; MACE = major adverse cardiovascular events.

## Data Availability

Data are taken from literature.

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
