# Peer review of "Myocardial Work by Echocardiography: Principles and Applications in Clinical Practice"

_jcm, 2021, doi:10.3390/jcm10194521_

Round 1

Reviewer 1 Report

This review paper by Ilardi et al reports on recent findings in the field of myocardial work (MW) calculation. While providing the reader with an overview of references to manuscripts that have tested MW indices in a variety of cardiac pathologies, the current review also contains important limitations. These include:

  1. The review contains many strong statements which are not very substantiated. A review should – ideally – be a place where conclusions from original research papers are put in context, rather than a direct copy of statements made in those papers.
  2. Most of the figures provided are not sufficiently discussed. The authors provide examples of pressure-strain loops (PSL) – which is good, because often forgotten – but do not detail why certain PSL shapes lead to specific differences in MW indices. Since PSL shape forms the basis of the MW indices, this should be primordial in a review paper on MW.
  3. An important limitation of EchoPAC’s MW algorithm goes back to its core calculation: its LV pressure reference curve is deducted from invasive data from patients with LBBB (see ref. 12 from Russell et al). It is not discussed in the current review how this would affect measurements in the reported plethora of cardiac pathologies.
  4. MW indices are not load-independent. They are – at most – only less load dependent than myocardial strain. In fact, MW is strongly dependent on myocardial strain, and only multiplies it with a factor, i.e. LV pressure. Most of the reported findings in MW are therefore a direct translation of the findings in myocardial strain. This deserves attention in the review.

Minor comments

  1. Most of the labelling of the used bulls eyes are hard to read. Perhaps increase the size/resolution?
  2. Figure 1. MW indices in EchoPAC do not use peak systolic strain. Strain is considered during a large portion of the cardiac cycle, from MVC to MVO.
  3. The used references in the field of CRT require an update and/or a more balanced choice of cited papers. Furthermore, the Prospect trial (ref. 17) can hardly be called “recent” anymore.
  4. Typo’s. Page 1: dis-eases. Page 2: Rus-sell; de-rived. Page 3: PS instead of PSL in the second to last paragraph. Page 6: extra space before second sentence. Page 11: pre-diction; and extra space in the first sentence of the last paragraph.

Reviewer 2 Report

The present manuscript reviews the principles and main applications of the newer defined myocardial work, an echocardiographical estimate of ventricular work based on global longitudinal strain and brachial arterial blood pressure. The paper is useful particularly for researchers in cardiovascular pathophysiology, as this parameter at the moment is generally not used in the daily practice.

I have some major points that I believe the authors should address in this paper:

  1. EF is of course load dependent, as the vast majority of parameters estimating left ventricular function. Intrinsic contractility is not measurable by current techniques. However, EF ´s major limitation is its very low sensitivity in detecting declining ventricular function. The Starling effect has its role in earlier stages of disease (the "compensated" phase) and not in more advanced, "decompensated" phases. I suggest the authors to rephrase the paragraph on these aspects  in Introduction, page 1.
  2. Many other functional parameters have better predictive power than EF, in a variety of pathologies. What is interesting for the readers to know is how much additional predictive power MW has to both GLS, BP, and the left ventricular ejection time. By the nature of the statistics involved, each parameter composed by several other parameters, will have an incremental power to predict events to each of its components taken separately. So, only a significant increase in its predictive capabilities would be relevant, both in research and in clinical practice.
  3. In this context, I would like the authors to discuss particularly the papers in which the incremental power of MW has been addressed properly, and to objectify this by putting a number/ a % increase. 
  4. In this line, the authors can also specify in which papers the additive value of calculating MW in addition to GLS, BP and ejection time, has not been discussed. This can be pointed out as a new area of research.
  5. I miss a more critical approach towards MW. GLS ´s limitations are inherited by MW. GLS is very sensitive to image quality and should only be applied on good quality loops (recognizing that this can at the moment not be quantified objectively). Apical foreshortening is also a major limitation causing faulsely increased GLS. Please address this in limitations.
  6. In figure 1, both these weaknesses of GLS are visible. The images have a degree of foreshortening, and the quality, as it appears in the reproduced picture, is not ideal. Please replace the apical images with another acquisition.
  7. The value of MW in hypertension and patients with stiff arteries should be addressed more in detail, especially since the formula includes a peripheral BP measurement.

Round 2

Reviewer 1 Report

I read with interest the revision prepared by the authors. I appreciate the changes they implemented. However, I do remain with comments, mainly with regard to the still lacking critical point-of-view that a review paper should attain.

Major comment:

  • Would the authors be willing to detail where they revised the manuscript in order to create more context for the findings of the original research papers which are cited? I see little added discussion or new critical considerations.

Minor comment:

  • Referring to the subsection on CRT. The newly added references to manuscripts which focused solely on strain add little to a review article dedicated to myocardial work. Some help for the authors to guide them to more recent papers on myocardial work in large(r) patient cohorts: doi: 10.1093/ehjci/jeaa003 + doi: 10.1093/eurheartj/ehaa603.

Author Response

I read with interest the revision prepared by the authors. I appreciate the changes they implemented. However, I do remain with comments, mainly with regard to the still lacking critical point-of-view that a review paper should attain.

Major comment:

  • Would the authors be willing to detail where they revised the manuscript in order to create more context for the findings of the original research papers which are cited? I see little added discussion or new critical considerations.

We thank the reviewer for the suggestion. We implemented the discussion on the findings described in the paper, adding new critical considerations in the following paragraphs:

  • Pag 7 lines 215-223
  • Pag 8 lines 235-245
  • Pag 10 lines 308 – 310
  • Pag 13 lines 412 - 415

Minor comment:

  • Referring to the subsection on CRT. The newly added references to manuscripts which focused solely on strain add little to a review article dedicated to myocardial work. Some help for the authors to guide them to more recent papers on myocardial work in large(r) patient cohorts: doi: 10.1093/ehjci/jeaa003 + doi: 10.1093/eurheartj/ehaa603.

According to this comment, we added the two suggested papers (ref 24 and 25) and discussed their results in the text  (pag 7 lines 201-209).